# Spectroscopic Evaluation of Red Blood Cells of Thalassemia Patients with Confocal Microscopy: A Pilot Study

**DOI:** 10.3390/s20144039

**Published:** 2020-07-21

**Authors:** Laura Rey-Barroso, Mónica Roldán, Francisco J. Burgos-Fernández, Susanna Gassiot, Anna Ruiz Llobet, Ignacio Isola, Meritxell Vilaseca

**Affiliations:** 1Centre for Sensors, Instruments and Systems Development, Technical University of Catalonia, 08222 Terrassa, Spain; francisco.javier.burgos@upc.edu (F.J.B.-F.); meritxell.vilaseca@upc.edu (M.V.); 2Unit of Confocal Microscopy, Service of Pathological Anatomy, Pediatric Institute of Rare Diseases, Hospital Sant Joan de Déu, 08950 Esplugues de Llobregat, Spain; mroldanm@sjdhospitalbarcelona.org; 3Institute of Pediatric Research, Hospital Sant Joan de Déu, 08950 Esplugues de Llobregat, Spain; sgassiot@sjdhospitalbarcelona.org (S.G.); imisola@sjdhospitalbarcelona.org (I.I.); 4Laboratory of Hematology, Service of Laboratory Diagnosis, Hospital Sant Joan de Déu, 08950 Esplugues de Llobregat, Spain; 5Service of Pediatric Hematology, Hospital Sant Joan de Déu, 08950 Esplugues de Llobregat, Spain; aruizl@sjdhospitalbarcelona.org

**Keywords:** red blood cells, thalassemia, spectroscopy, confocal microscopy, spectral imaging, autofluorescence

## Abstract

Hemoglobinopathies represent the most common single-gene defects in the world and pose a major public health problem, particularly in tropical countries, where they occur with high frequency. Diagnosing hemoglobinopathies can sometimes be difficult due to the coexistence of different causes of anemia, such as thalassemia and iron deficiency, and blood transfusions, among other factors, and requires expensive and complex molecular tests. This work explores the possibility of using spectral confocal microscopy as a diagnostic tool for thalassemia in pediatric patients, a disease caused by mutations in the globin genes that result in changes of the globin chains that form hemoglobin—in pediatric patients. Red blood cells (RBCs) from patients with different syndromes of alpha-thalassemia and iron deficiency (including anemia) as well as healthy (control) subjects were analyzed under a Leica TCS SP8 confocal microscope following different image acquisition protocols. We found that diseased RBCs exhibited autofluorescence when excited at 405 nm and their emission was collected in the spectral range from 425 nm to 790 nm. Three experimental descriptors calculated from the mean emission intensities at 502 nm, 579 nm, 628 nm, and 649 nm allowed us to discriminate between diseased and healthy cells. According to the results obtained, spectral confocal microscopy could serve as a tool in the diagnosis of thalassemia.

## 1. Introduction

Red blood cells (RBCs) are specialized cells in charge of oxygen transportation throughout the body. They contain a tetramer called hemoglobin that is able to bind oxygen and carbon dioxide molecules. Healthy RBCs have a biconcave shape with an average volume of 90 to 95 μm^3^ and great deformation capacity when passing through capillaries of the circulatory system. Due to defects in genetic information coding for RBCs, their shape, number, and deformability can be altered, as well as their capacity to transport oxygen in the blood. In thalassemia, mutations in the globin genes result in a quantitative reduction in the rate of synthesis or the absence of globin chains which form hemoglobin (alpha or beta-thalassemia) [1]. Clinical manifestations of thalassemia syndromes range from no symptoms in asymptomatic carriers to serious abnormalities that include severe anemia, extramedullary hematopoiesis, skeletal and growth deficits and iron overload, with a significantly shortened life expectancy in the absence of treatment. The severity of the clinical features correlates with the number of functioning globin genes that are lost [2]. Iron deficiency is the other major cause of microcytic anemia besides thalassemia, and this can be associated with a markedly abnormal RBC morphology, including hypochromic/microcytic cells, as in the more severe forms of thalassemia.

The diagnosis of thalassemia is based on RBC morphology under conventional optical microscopy and RBC indices, which consist of estimating the levels of adult hemoglobin A1 (HbA), its variant hemoglobin A2 (HbA2), and fetal hemoglobin (HbF), which are most frequently determined by automated high-performance liquid chromatography [3]. These techniques are sometimes not sufficiently specific enough to distinguish between mild and minor forms of the disease, which show no symptoms or very mild symptoms and have similar blood hemoglobin and corpuscular volume values; additionally, thalassemia can also be confused with other causes of anemia such as iron deficiency. Therefore, complex and expensive genetic studies are often required to diagnose individuals with this disease, because although they remain asymptomatic, even if the parents have a mild form, their offspring can be affected with severe forms.

Authors have analyzed RBCs under experimental and commercial spectroscopic systems with the aim of overcoming the limitations of traditional techniques. To date, research studies have attempted to establish reflectance, emission and/or absorption differences in the ultraviolet (UV), visible (VIS) and near infrared (NIR) ranges between healthy and unhealthy RBCs. These techniques were shown to be useful for automatizing blood cell classification since they showed different spectral signatures. Akuwudike et al. [4] used spectral absorbance to distinguish sickle cell hemoglobin from normal adult hemoglobin, evaluating the range from 250 nm to 650 nm. Gunasekaran et al. [5] proved that there are statistically significant optical density differences in the UV and VIS ranges (200–700 nm) between RBCs of patients with leukemia, anemia, liver cirrhosis, thalassemia and diabetes with respect to healthy individuals. To better understand the variability and determinants of oxygen affinity on a cellular level, Di Caprio et al. [6] designed a microfluidic chip to illuminate RBCs with blue and red light-emitting diodes (LEDs). Blue light was used to differentiate between oxygenated and deoxygenated hemoglobin, and red light was employed for cell volume quantification. Liu et al. [7] proved that the infrared (IR) response revealed changes in the secondary structure of hemoglobin from beta-thalassemic patients compared to controls. Alsalhi et al. [8] used a spectrofluorometer for the spectral detection of thalassemia in a preliminary study. They observed two peaks in the autofluorescence emission of RBCs at 580 nm and 630 nm, mostly due to the basic and neutral forms of porphyrin—a type of nitrogenous biological pigment that forms the heme group. In order to better discriminate between thalassemia and iron deficiency anemia, Devanesan et al. [3] identified and quantified a certain set of fluorescent metabolites in blood samples of subjects affected by these diseases. Nevertheless, in all of the aforementioned studies, the spectral differences that may underlie different clinical presentations of thalassemia (especially mild forms) were not reported. This might be due to a lack of spectral differences, in the case of asymptomatic forms, or because none of these studies made use of imaging sensors; therefore, the integrated spectroscopic information provided is not sufficiently spatially accurate to reveal such differences.

The use of spectral imaging improves RBC analysis since it adds spatial resolution to spectroscopic data. Dai et al. [9] used a molecular hyperspectral imaging system to identify blood cells. This system was developed according to a push-broom approach and covered the VIS and NIR ranges (400–860 nm); spectral pattern traits and similarity measures were obtained from the background, red cells, lymphocytes, nuclei, and plasma of tumor cells. Conti et al. [10] made use of hyperspectral dark field microscopy to measure the scattering spectra of RBCs from 400 nm to 1700 nm. The analyses were carried out with a hyperspectral microscope equipped with a high-power halogen lamp, and eight spectral signatures were identified for RBC molecular components. Kurtuldu et al. [11] developed a hyperspectral microscope based on a liquid crystal tunable filter to analyze RBCs and detect by image classification the different elements in a cell from 420 nm to 730 nm. Robison et al. [12] designed a snapshot hyperspectral system capable of capturing several spectral bands simultaneously (419–494 nm) by combining a commercial snapshot system and a microscope. They found that white blood cell features were most prominent in the 428 nm to 442 nm band. 

Other authors have analyzed RBCs under confocal microscopy, which, unlike traditional microscopy, allows the sample to be scanned at different depths. Axial cuts of the sample contain autofluorescence or reflectance information as well as the actual morphology of RBCs, which could have numerous potential applications in the diagnosis of RBC diseases. For instance, reflectance confocal microscopy (RCM) has been demonstrated to be useful for obtaining information about the physiological properties of RBCs at high resolution and without the need for fluorescence labeling [13,14]. Golan et al. [13] combined RCM with flow cytometry, taking advantage of the unidirectional flow of blood within small capillary vessels. Zeidan et al. [14] simulated and obtained actual reflectance confocal images of RBCs in combination with flow cytometry to study their morphology and physiology. Khairy et al. [15] obtained tridimensional (3D) confocal microscope images of RBCs labeled with fluorescent dyes and compared them to mathematical simulations of the shapes by means of spherical harmonic series expansions. Rappaz et al. [16] compared the morphological values obtained from different imaging techniques, including confocal microscopy. In order to perform a volume assessment of RBCs under confocal laser scanning microscopy (CLSM), cells were labeled with a fluorescent dye and excited at 561 nm, and the emission was collected from 580 nm to 700 nm. Lima et al. [17] made use of confocal microscopy in a more innovative way to measure individual RBC motions through micro-vessels by combining the former imaging technique with a particle tracking velocimetry system. Yakimov et al. [18] used fluorescence lifetime imaging microscopy, an imaging modality implemented in confocal microscopy, to determine the biochemical state of white blood cells provided by native fluorophores.

As can be inferred from the state-of-the-art approaches, the use of spectral imaging techniques together with CLSM could provide simultaneous spectroscopic and 3D information about the composition of functional molecular complexes or substances that characterize the metabolic state of RBCs. Therefore, this approach can be a potential tool in diagnosing RBC diseases, providing more data beyond an accurate representation of their morphology, as spectral information would also be available. Then, RBCs could be studied by building 3D maps of the functional information of the cell, collecting the reflectance, autofluorescence, or even fluorescence arising from the staining of different cellular components with the use of extrinsic fluorescent probes. 

The purpose of the present work is to analyze the spectral and morphological characteristics of healthy and diseased RBCs—specifically RBCs from pediatric patients with thalassemia, under a spectral confocal microscope, which is a powerful imaging instrument that has not yet exploited as a diagnostic tool for RBC diseases. Spectral confocal microscopy allows the sequential study of the cell structure and molecular components, which is very helpful in diagnosing RBC diseases, where the isolated assessment of the cell shape is insufficient. In addition, blood samples from patients with iron deficiency are also studied and compared to thalassemia samples. 

## 2. Materials and Methods

### 2.1. Sample Preparation 

The management of blood samples in confocal microscopy is somewhat special and has not often been described in the literature. Therefore, in this work, we had to establish a protocol for loading blood samples into the microscope, carefully choosing the container dish, anticoagulant agents, the time for the samples to stabilize and the conditions inside the microscope cabin to perform live imaging. Living RBCs are delicate structures that need to remain under the same conditions as inside the human body in order to preserve their shape and molecular complexes. Therefore, neither the addition of solvents, such as saline solution, nor the centrifugation of the sample to remove other cellular types and components in blood was considered. Nevertheless, it was necessary to acquire spectral image acquisition sequences over several areas or fields to reduce the variability that could be found within the sample. This made the duration of the experiments to be relatively long so in order for RBCs to remain unaltered, the addition of two frequently used anticoagulant agents, lithium heparin and ethylenediaminetetraacetic acid (EDTA), was considered to avoid sample corruption. Lithium heparin was found to be technically more suitable for the experimental assays, preserving the optical properties of RBCs for a longer time. 

Blood samples from 17 pediatric patients were evaluated, comprising 12 males and five females between 1 and 17 years old, including patients with different forms of alpha-thalassemia, patients with different degrees of iron deficiency and healthy individuals as controls. Samples were not labeled with extrinsic fluorescent probes to avoid overlying spectral information. Since living cells move inside their medium, the use of dishes with no coating to load samples would have made the selection of cells during image post-processing significantly more difficult. Therefore, CELLview™ cell culture dishes with four compartments (Greiner Bio One GmbH, Courtaboeuf, France), which incorporate a cell-adhesion layer preventing the movement of cells during measurement, were used. Blood samples were collected in tubes containing lithium heparin. RBC indices were analyzed using an ADVIA 2120i hematology analyzer (Siemens Healthcare Diagnostics Inc., Erlangen, Germany) within 2 hours after blood collection. 

The study group consisted of eight healthy subjects, labeled TC1–TC8, including one patient with HbH alpha-thalassemia (homozygous for a HbA2 c.*93_*94delAA mutation), which is considered severe, labeled T1; a patient with alpha-thalassemia minor (Southeast Asian [SEA] heterozygous deletion), identified as T2; and four asymptomatic alpha-thalassemia carriers (3.7 kb heterozygous deletion), labeled T3–T6. In addition, the samples of the three patients suffering from different degrees of iron deficiency were analyzed and labeled TA1–TA3. Samples were characterized in terms of the amount of hemoglobin in the blood (Hb) in grams per deciliter (g/dL); the mean corpuscular volume (MCV) in femtoliters (fL), which indicates the average size of RBCs; and mean corpuscular hemoglobin (MCH) in picograms (pg), which quantifies the amount of hemoglobin per RBC. These RBC indices are presented in Table 1 along with the average values for each study group. Patient T1 was evaluated again one year after the initial analysis; these data are labeled T1_2.

Approximately 200 μL of each blood sample was introduced in an adherent Petri dish and mounted inside the cabin in the microscope for temperature and CO_2_ control in live imaging.

### 2.2. Spectral Confocal Imaging

The samples were analyzed under a Leica TCS SP8 confocal microscope with stimulated emission depletion (STED) at 3× super resolution (Leica Microsystems GmbH, Mannheim, Germany), equipped with a detection unit that allowed spectral discrimination using hybrid detectors. These detectors are capable of detecting signals arising from RBCs from 400 nm to 790 nm. The microscope incorporates two lasers for excitation, a diode laser with an emission of 405 nm and a white laser that emits from 470 nm to 670 nm, combined with an acoustic-optic tunable filter (AOTF). The imaging acquisition protocol followed for the confocal microscope to collect spectral traits of RBCs corresponded to the emission (fluorescence) configuration. Preliminary experiments on the reflectance configuration did not show differences between diseased and healthy cells, and consequently this analysis was dropped.

To collect the spectral emission of RBCs produced by autofluorescence, a 63× (NA 1.4, oil) plan-apochromatic objective was used. The confocal microscope could focus at several depths within the volume of blood, where RBCs were clearly differentiated and displaced throughout the sample, acquiring several fields to evaluate emission uniformity. The image format chosen was 400 pixels × 400 pixels using a four airy unit (AU) pinhole. Samples were excited at 405 nm with a blue diode laser line and the AOTF was set at 65%. Emission images from 425 nm to 790 nm were acquired with a spectral window of 20 nm and in steps of 7 nm in a so-called xyλ scanning sequence. The variation in intensity of a particular spectral component, encoded using 8 bits, was represented on the screen using a pseudo-color look-up table. Emission detection was performed at a galvanometric bidirectional speed of 1000 Hz to avoid artefacts of cell movement and at the same time provide sufficient exposure time to excite the sample at each point. The choice of wavelength can be critical in cell discrimination. Greiner et al. [19] proved in a flow cytometry study that better discrimination between RBC and white blood cell populations could be obtained at a shorter wavelength of 413 nm compared to 488 nm. Similarly, the excitation wavelength of a 488 nm argon laser was also tested in this work to excite the samples. However, no difference other than lower-intensity emissions compared with the 405 nm excitation was observed. Accordingly, only the results from the experiments with excitation at 405 nm are described below. The high sensitivity of the hybrid detectors allowed us to amplify the low autofluorescence emission of RBCs and reduce the noise due to their higher dynamic range.

Data from all studies were analyzed using the Leica Application Suite X (LAS X) software (Leica Microsystems GmbH, Mannheim, Germany). This software allows the determination of spectral signatures of different regions of interest (ROIs) within the imaged area of the sample (field) and displays the mean intensity of all pixels for each ROI versus the wavelength. Between 10 and 20 areas or fields over each sample were studied. Mean values of the selected ROIs within a field were obtained. Numerical data were exported to a spreadsheet (Excel 2000; Microsoft Corp., Redmond, WA, USA). The mean spectral emission intensity and standard error were calculated for all fields examined and were expressed in arbitrary units (a.u.). The significance of differences in terms of the spectral values collected at several wavelengths (from 425 nm to 790 nm in steps of 7 nm) were compared among controls, patients with thalassemia and patients with anemia (by pairs) using an unpaired Student’s t-test with statistical significance set at *p* = 0.05 (IBM SPSS Statistics for Windows, version 24.0; IBM Corp., Armonk, NY, USA).

## 3. Results and Discussion

Figure 1 contains sequences of confocal emission images captured at different wavelengths for some samples when excited at 405 nm: TC1 in the control group; T1 and T2 corresponding to individuals with HbH alpha-thalassemia (severe) and SEA alpha-thalassemia (mild), respectively; and TA3, corresponding to iron-deficiency anemia. For each sample, spectral emission images at 453 nm, 502 nm, 579 nm, 628 nm, and 649 nm of a given imaged field are included, as well as the average emission curves considering the cellular structures (RBCs, surrounded by a dashed white contour, and neutrophils) that appeared to emit differently to the entire background of the samples.

It can be seen that, in all instances, there is a peak around 502 nm created by all sample emission, background and cellular structures. Paying attention to the spectral images collected at 579 nm, 628 nm and 649 nm, it can be inferred that, for samples corresponding to HbH disease (T1), SEA deletion (T2) and iron deficiency (TA3), some cellular structures appear brighter. Sometimes, one of the latter peaks of emission is greater than that at 502 nm (T1 and TA3). Specifically, in the case of iron-deficiency anemia, the cells that appear brighter in the spectral image at 579 nm contribute to the average emission curve, providing a prominent peak at this wavelength.

Emission images were captured for different areas of the samples where a sufficiently large number of RBCs were visible. These cells were experimentally determined to emit differently than the rest of the structures in the background, emitting more intensively at the longest wavelengths. The populations of these cells were found to be especially high in the samples of the diseased groups (HbH thalassemia, SEA thalassemia, thalassemia carrier and iron deficiency). In order to obtain a general overview of the emission, circular regions of 4 μm were sketched on top of one to 10 RBCs as ROIs on every field, depending on whether the sample had a higher or lower population number over which the sequences of confocal images were captured. The selection of round-shaped neutrophils was avoided, which were identified experimentally for their shape, larger size and very bright emission at 502 nm. Only spectral sequences in which cellular movement was relatively low were included in the ROI emission analysis. Finally, T4 and TC2 were not included due to the insufficient number of evaluated fields for their analysis.

The specific emission intensity values from 450 nm to 649 nm of some representative subjects are included in Table 2; Figure 2 depicts the corresponding spectra as the average emission autofluorescence of the aforementioned ROIs. In the spectrum, a common emission peak was observed around 502 nm, although in some samples this could be shifted some nanometers above or below. In addition, samples T1 (HbH), T2 (SEA heterozygous deletion) and all samples corresponding to iron deficiency (TAs) showed higher emissions around 628 nm and 649 nm. The intensity of these three peaks was different for all samples analyzed. Moreover, we observed that the number of RBCs with a different spectrum than the background varied from one sample to another; control samples presented the most uniform emission within blood structures.

Studies using the state-of-the-art approaches have obtained similar emission spectra for normal and diseased RBCs. We excited RBCs at around 400 nm, but according to Zheng et al. [20], hemoglobin can also be two-photon excited at around 600 nm, and a very similar emission spectrum for healthy RBCs to the one described in this work would be obtained. The common peak around 500 nm might be produced by lipofuscins, lipofuscin like-lipids or retinoids, whose autofluorescence in terms of their spectral shape and amplitude is related to senescence and oxidation degree [21]. Distinct and well-defined bands beyond 600 nm are generally attributed to porphyrins and a heme-altered metabolism. For instance, Liu et al. [7] detected a peak around 628 nm produced in the emission curves of some samples, which was attributed to porphyrin. This compound is a nitrogenized biological pigment whose derivative products include hemoproteins, which are made of a combination of porphyrin, metals and proteins. Porphyrin provides RBCs with their characteristic red color. It is thought that the amount of free porphyrin in blood is greater in patients suffering from alpha-thalassemia and iron-deficiency anemia than in healthy individuals [22], which is consistent with the findings of our study. Stockman et al. [23] proved that free porphyrin in blood with iron-deficiency anemia was even higher. Patients with protoporphyria also present similar spectra to thalassemia, with the most prominent emission peak around 630 nm [24]. 

In order to quantify this concept, the ratio between the intensity at the first and second emission peaks of 502 nm and 628 nm was calculated, as well as the ratio between peaks at 502 nm and 649 nm. In addition, in order to account for the number of RBCs that were found to emit with a special spectrum, selected as ROIs on each sample, we multiplied the ratios by the average number of cells per field found for each, as follows:Ratio 1 = (I_502_/I_628_)·*RBC*,(1)
Ratio 2 = (I_502_/I_649_)·*RBC*,(2)
where I_λ_ is the intensity at different wavelengths and *RBC* is the average number of RBCs per field.

Table 3 shows that the higher the severity of alpha-thalassemia, the larger the calculated rates. Mild and minor alpha-thalassemia individuals (T2, T3, and T6) showed intermediate values which were slightly higher than those of control patients, while HbH alpha-thalassemia (T1 and T1_2) and iron-deficient patients (TA1–TA3) in addition to T5—an alpha-thalassemia carrier—presented higher intensity ratios. Therefore, a higher ratio seems to be associated with a greater severity of anemia. If an additional ratio between the intensity recorded at 502 nm and 579 nm (prominent for iron deficiency) is taken into account (Equation (3)), it becomes possible to differentiate iron-deficiency samples from thalassemia samples (Table 3).
Ratio 3 = I_502_/I_579_(3)

In the statistical analysis in terms of ratio 1, (I_502_/I_628_)·*RBC*, the student’s t-test showed significant differences between control patients and those with thalassemia (*p* = 0.01) and with anemia (*p* = 0.002), whereas it yielded smaller, but still significant, differences between individuals with thalassemia and anemia (*p* = 0.045). Similar results were found in terms of ratio 2, (I_502_/I_649_)·*RBC*, although in this case differences between patients with thalassemia and anemia were not significant: for control–thalassemia, *p* = 0.001; for control–anemia, *p* = 0.001, and for thalassemia–anemia, *p* = 0.120. 

On the other hand, the use of ratio 3, I_502_/I_579_, only provided statistically significant differences between patients with thalassemia and anemia (*p* = 0.048), whereas the comparisons between control individuals and those with thalassemia (*p* = 0.718) and with anemia (*p* = 0.243) were not significant. 

Figure 3 graphically represents the results of Table 3; it can be seen that the different diseases are differentiated for the two first calculated ratios (Figure 3a), whereas the third ratio (Figure 3b) computed between 502 nm and 579 nm, is useful to discriminate samples corresponding to iron deficiency from the others (except in one case).

## 4. Conclusions

The combination of CLSM with spectrofluorometry techniques is a powerful tool that allows the direct analysis of global and unique fluorescent pixels and their 3D location ex-vivo in the whole specimen, minimizing the artefacts associated with sample processing; other techniques, such as atomic force microscopy [25], may allow the RBC membrane to be inspected in detail but cannot retrieve images at any desired thickness of the specimen. Our approach also allows RBC discrimination with particular endogenous fluorescence signals. Other techniques, such as phase imaging by interferometry, which is often used to quantify changes in RBCs, may allow the imaging of the specimen at specific depths from which the signal of interference arises, but they do not provide sufficient wavelengths of excitation as well as spectral windows of detection to explore the different spectral traits in depth [26]. Since thalassemia has unspecific imaging characteristics, of the few described state-of-the-art approaches that we tried to reproduce, the best tool to work with is confocal imaging and the super-resolution provided by the microscope used. The descriptors (I_502_/I_628_) · *RBC* and (I_502_/I_649_) · *RBC* (ratios 1 and 2) related to the intensities measured at 502 nm, 628 nm and 649 nm when exciting RBCs at 405 nm allowed a discrimination between healthy and diseased individuals that presented with anemia (thalassemia or iron deficiency) and between different degrees of influence in alpha-thalassemia, with less accuracy due to the small sample size. Ratio 3, I_502_/I_579_, differentiates most iron deficiencies from thalassemia, although additional samples should be analyzed to validate its performance. The difference in fluorescence resulting from these parameters may reside in heme group degradation, which is associated to oxidative stress, as described by some of the authors in the literature [27]. Nagababu et al. [28] have found heme degradation products in thalassemic mice, which share with humans the gene clones that might be affected in this and other hemoglobinopathies. It is interesting that, in this case, instead of exciting at 405 nm, the authors excited samples with a 321 nm laser wavelength and obtained two fluorescent emission bands, with a predominant peak at 480 nm. It would have been interesting to excite our samples with shorter wavelengths than 405 nm without compromising the viability of the cells; however, with commercial confocal microscopes, this is the shortest wavelength available. In addition, the authors suggested that the cell membrane might be affected due to the release of iron from the heme group degradation, with an increase in immunoglobin G antibody binding. Other authors have also suggested that the function of the cell membrane might be affected in some ways; for instance, membrane skeletal spectrin is thought to be related to the amount of hemoglobin [29]. There is an imbalance in the bilipid membrane affecting the cytoadherence; thus, these diseased RBCs have difficulties circulating through certain vessels—for instance, the ones in the spleen, causing splenomegaly. Therefore, it would be interesting to carry out another type of assay, referred to as the immuno-labeling of the cell membrane, which can only be carried out with the cells fixated; this would be helpful for detecting affectations in the structure of the cell membrane, which are not currently clearly determined. Nevertheless, another advantage of the combined use of the imaging techniques described in this work, unlike other classical tests (such as complete blood count, blood smears, or electrophoresis), is that it does not necessarily require reactives or the use of markers and fixating substances used to capture absorption spectra [30]. Thus, tests can be carried out in normal physiological conditions without the need to prepare hemolyzed serums and with a relatively low volume of RBCs.

Another conclusion that we have drawn is that the establishment of a diagnosis protocol in hospitals that include a pathological anatomy service would be relatively easy, since the healthcare circuit for hematology samples to be directly analyzed in a Confocal Microscopy Unit works efficiently. In addition, a diagnosis protocol based on confocal imaging has been compared to the genetic studies that are currently used and will certainly shorten the response time and reduce the costs of the analyses.

Future work will consist of expanding the set of samples to corroborate the effectiveness of the ratios described. We will also need to corroborate the results by trying to reproduce the same experiments in a different hospital. Open source neural network models will be used to analyze our set of images in further detail, and thus we will obtain better screening results. The current research and next experiments may offer hematologists with a new approach to improving diagnostic strategies and searching for involved genes.

## Figures and Tables

**Figure 1 sensors-20-04039-f001:**
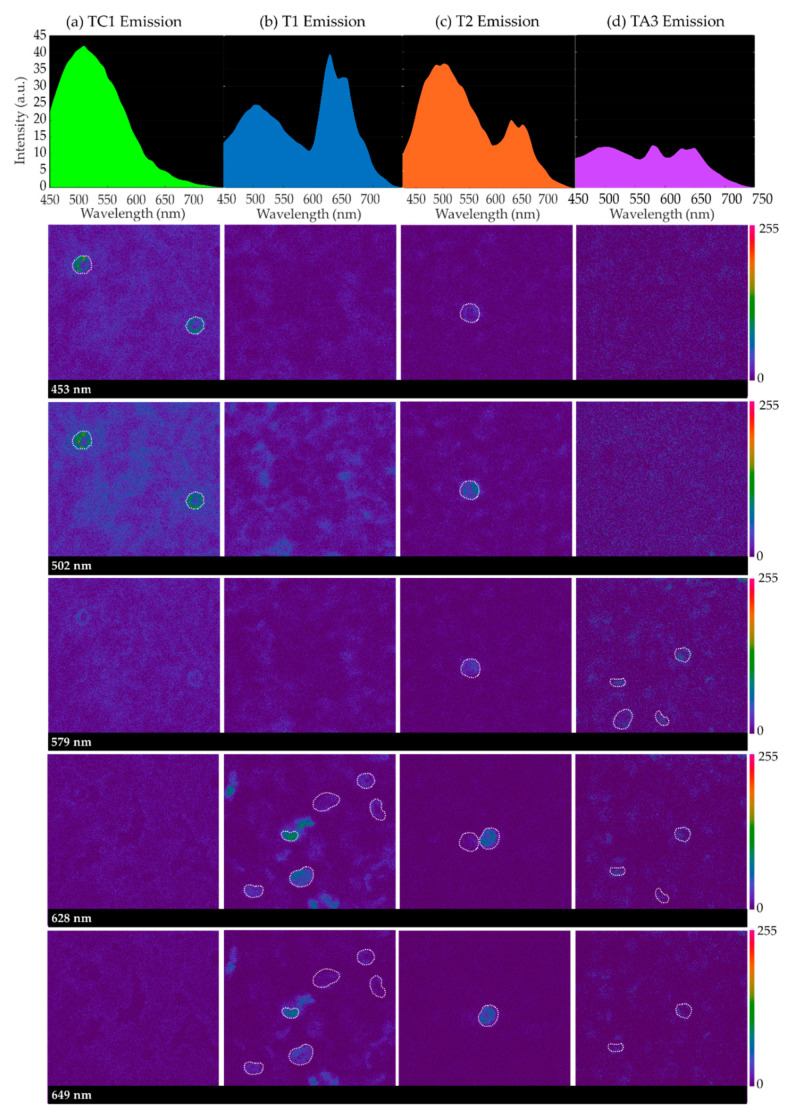
Intensity vs. wavelength (top) and images from autofluorescence of red blood cells (RBCs) (bottom) corresponding to 453 nm, 502 nm, 579 nm, 628 nm, and 649 nm wavelengths, for the following samples: (**a**) TC1, from a control patient; (**b**) T1, HbH (severely diseased); (**c**) T2, SEA heterozygous deletion; (**d**) TA3, from a patient with iron-deficiency anemia. It can be appreciated that cellular structures appear brighter in T1, T2, and TA3.

**Figure 2 sensors-20-04039-f002:**
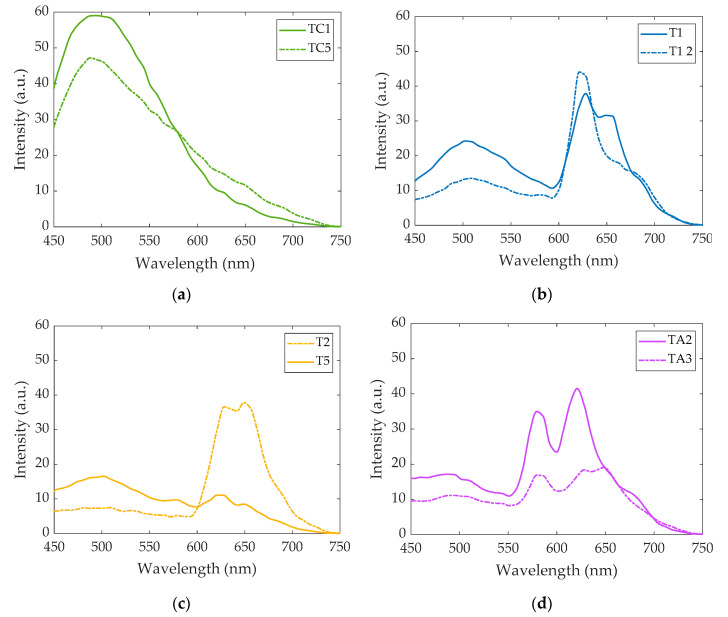
Intensity vs. wavelength (average curves) computed from autofluorescence images of samples (**a**) TC1 and TC5 (healthy subjects); (**b**) T1 and T1_2 (HbH disease); (**c**) T2 (SEA heterozygous deletion) and T5 (3,7 kb heterozygous deletion); (**d**) TA2 and TA3 (iron deficiency).

**Figure 3 sensors-20-04039-f003:**
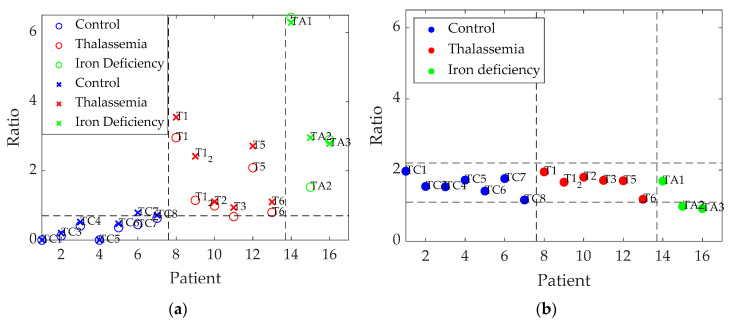
Graphical representation of averaged ratios of main peaks of emission. (**a**) Circles represent values of ratio 1, (I_502_/I_628_)·*RBC*, and X’s correspond to ratio 2, (I_502_/I_649_)·*RBC*. (**b**) Ratio 3, I_502_/I_579_.

**Table 1 sensors-20-04039-t001:** Indices for control, thalassemic, and iron-deficient patients: hemoglobin (Hb) in g/dL; medium corpuscular volume (MCV) in fL; medium corpuscular hemoglobin (MCH) in pg.

Indices	Hb (g/dL)	MCV (fl)	MCH (pg)
TC1	14.1 ± 0.32	87 ± 0.71	29.4 ± 0.42
TC2	11.8 ± 0.49	80 ± 1.77	26.9 ± 0.46
TC3	13.3 ± 0.03	89 ± 1.41	29.5 ± 0.46
TC4	13.7 ± 0.18	85 ± 0.00	28.5 ± 0.11
TC5	13.6 ± 0.14	83 ± 0.71	28.0 ± 0.07
TC6	13.3 ± 0.03	80 ± 1.77	26.7 ± 0.53
TC7	12.3 ± 0.32	88 ± 1.06	28.6 ± 0.14
TC8	13.5 ± 0.11	86 ± 0.35	28.2 ± 0.00
Mean	13.2 ± 0.00	85 ± 0.00	28.2 ± 0.00
T1	7.8 ± 1.17	68 ± 1.51	16.8 ± 2.08
T1_2	9.0 ± 0.72	75 ± 1.13	18.2 ± 1.55
T2	10.8 ± 0.04	63 ± 3.40	19.4 ± 1.09
T3	10.4 ± 0.19	74 ± 0.76	25.1 ± 1.06
T4	14.3 ± 1.29	74 ± 0.76	25.4 ± 1.17
T5	11.3 ± 0.15	75 ± 1.13	25.4 ± 1.17
T6	12.6 ± 0.64	78 ± 2.26	25.6 ± 1.24
Mean	10.9 ± 0.00	72 ± 0.00	22.3 ± 0.00
TA1	9.9 ± 0.69	81 ± 4.04	25.0 ± 0.98
TA2	12.9 ± 1.04	77 ± 1.73	26.0 ± 1.55
TA3	10.6 ± 0.29	65 ± 5.19	18.9 ± 2.54
Mean	11.1 ± 0.00	74 ± 0.00	23.3 ± 0.00

**Table 2 sensors-20-04039-t002:** Average emission intensity (a.u.) at principal screening wavelengths and their associated standard error, depending on variability of the spectrum within the selected regions of interest (ROIs) and number of fields analyzed within the sample. Only data from some subjects of each group are shown.

Wavelength (nm)	450	502	579	628	649
TC1	42.0 ± 13.0	58.8 ± 13.5	26.5 ± 3.5	9.5 ± 1.2	6.2 ± 0.8
TC5	30.2 ± 6.8	46.0 ± 11.6	26.6 ± 6.7	14.7 ± 3.7	11.8 ± 3.1
T1	13.5 ± 1.9	24.2 ± 3.3	12.6 ± 1.4	37.9 ± 7.8	31.6 ± 4.1
T1_2	7.6 ± 1.3	13.3 ± 2.6	8.7 ± 1.3	42.9 ± 4.0	20.3 ± 2.1
T2	6.5 ± 0.3	7.3 ± 0.8	5.2 ± 1.1	36.6 ± 12.5	37.9 ± 5.1
T5	12.8 ± 0.4	16.6 ± 0.7	9.7 ± 0.7	11.1 ± 2.3	8.5 ± 2.0
TA2	16.0 ± 0.6	15.7 ± 0.6	35.0 ± 4.9	36.9 ± 2.5	19.1 ± 1.3
TA3	9.6 ± 0.2	10.9 ± 0.2	16.9 ± 1.2	18.4 ± 1.3	19.2 ± 1.3

**Table 3 sensors-20-04039-t003:** Averaged ratios for main emission peaks. I_λ_: intensity at different wavelengths; *RBC*: average number of RBCs per field.

Patient	Ratio 1 (I_502_/I_628_)·*RBC*	Ratio 2 (I_502_/I_649_)·*RBC*	Ratio 3 I_502_/I_579_
TC1	0.00	0.00	1.97
TC3	0.12	0.20	1.54
TC4	0.41	0.52	1.53
TC5	0.00	0.00	1.72
TC6	0.36	0.47	1.41
TC7	0.45	0.79	1.76
TC8	0.63	0.72	1.16
T1	2.96	3.55	1.95
T1_2	1.15	2.42	1.66
T2	0.99	1.11	1.80
T3	0.68	0.94	1.71
T5	2.09	2.72	1.70
T6	0.80	1.10	1.18
TA1	6.43	6.29	1.69
TA2	1.53	2.96	0.98
TA3	2.90	2.79	0.92

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
