# Peer review of "Spectroscopic Evaluation of Red Blood Cells of Thalassemia Patients with Confocal Microscopy: A Pilot Study"

_sensors, 2020, doi:10.3390/s20144039_

Round 1

Reviewer 1 Report

Manuscript Ref: sensors-843114
Review Report
Title: Spectroscopic evaluation of red blood cells of thalassemia patients with confocal microscopy: a pilot study

This manuscript describes an interesting work of the application of the spectral confocal microscopy that could serve as a tool in the diagnosis of thalassemia. The sensors-843114 is a very well written article, can do some good impact, and deserves to be published in the Sensors J.
Before publication, however, a number of improvements should be implemented:
• The abstract faithfully conveys the scope of investigations and conclusions drawn. The key words correspond well to the scope of the research.
• Lack of information about the statistical procedures
• add some new references to the conclusion section and also, I recommend to the authors to give and discuss in detail this section

Reviewer 2 Report

General comment

In this research, the authors explored the spectral confocal microscopy to evaluate red blood cells (RBCs) from health patients, and from patients with thalassemia or iron deficiency. The results demonstrate that it is possible to discriminate cases of thalassemia and iron deficiency. However, I see this study as preliminary, since it was restricted to a small group of patients. The manuscript is well written and organized. Thus, for me the manuscript can be accepted for publication after some minor corrections, especially, with an emphasis on the next steps that would be necessary to make this methodology feasible. Below I leave my specific comments.

Specific comments

1) To discriminate health and disease patients, and in this last case the type of hemoglobinopathy, the Principal component analysis (PCA) would be extremely useful. Have the authors tried to apply it?

2) The mean values of Table 1 should be accompanied by the respective standard deviations;

3) In Table 2, the authors say “Only data from some subjects of each group are shown.” Why not show it to everyone since there were only 17 patients?

4) What about the estimated cost of a diagnosis protocol based on the proposed spectral confocal microscopy sensing? Is it competitive?

5) The study of 17 pediatric patients is very narrow, and I think the authors would agree with me on that. Therefore, it would be interesting to mention the future steps that would be necessary to make this methodology feasible.

Reviewer 3 Report

Hemoglobinopathy is one of the common single-gene defect diseases in the world. In this work, the authors employed spectral confocal microscopy for the diagnosis of thalassemia in pediatric patients, and they analyzed the red blood cells from patients with different syndromes of alpha-thalassemia and iron deficiency (including anemia) as well as healthy (control) subjects under a Leica TCS SP8 confocal microscope. Although such a work is quite meaningful for the diagnosis of Hemoglobinopathy, I think the content is out of the Scope of Sensors, which mainly provides a forum for the science and technology of Sensors. This work may be more suitable published in Applied Spectroscopy.

Author Response

Dear Reviewer,

We would like to thank you for all the effort and time invested in reviewing the manuscript ID 843114 in its entirety:

‘Spectroscopic evaluation of red blood cells of thalassemia patients with confocal microscopy: a pilot study’

In the revised manuscript, we have taken into account the suggestions and followed most of the comments that you and the other reviewers have made.

We are glad that you think that our work can contribute to the improvement of the diagnosis certain hemoglobinopathies such as thalassemia and we humbly think that it does fit properly in Sensors’ special Issue “Color & Spectral Sensor”, since spectroscopic data has been acquired with techniques that include the use of complex sensors such as confocal microscopy and the fluorescence spectral images included take very relevant part on it.

The revised manuscript is now improved and we hope is appropriate to be published in Sensors’ under the section “Physical Sensors” and the special Issue “Color & Spectral Sensor”.

Yours sincerely,

Laura Rey-Barroso

Round 2

Reviewer 3 Report

The manuscript can be published although I think it is out of the scope of Sensors.